# Exploring the Interface between Inflammatory and Therapeutic Glucocorticoid Induced Bone and Muscle Loss

**DOI:** 10.3390/ijms20225768

**Published:** 2019-11-16

**Authors:** Justine M. Webster, Chloe G. Fenton, Ramon Langen, Rowan S. Hardy

**Affiliations:** 1Institute of Metabolism and Systems Research, University of Birmingham, Birmingham B15 2TT, UK; JMW801@student.bham.ac.uk (J.M.W.); CXF637@student.bham.ac.uk (C.G.F.); 2Centre for Endocrinology, Diabetes and Metabolism, Birmingham Health Partners, Birmingham B15 2TT, UK; 3Institute of Inflammation and Ageing, University of Birmingham, Birmingham B15 2TT, UK; 4Department of Respiratory Medicine, NUTRIM School of Nutrition and Translational Research in Metabolism, Faculty of Health, Medicine and Life Sciences, Maastricht University, 6211 LK Maastricht, The Netherlands; r.langen@maastrichtuniversity.nl; 5MRC Arthritis Research UK Centre for Musculoskeletal Ageing Research, University of Birmingham, Birmingham B15 2TT, UK

**Keywords:** glucocorticoid, muscle wasting, osteoporosis

## Abstract

Due to their potent immunomodulatory anti-inflammatory properties, synthetic glucocorticoids (GCs) are widely utilized in the treatment of chronic inflammatory disease. In this review, we examine our current understanding of how chronic inflammation and commonly used therapeutic GCs interact to regulate bone and muscle metabolism. Whilst both inflammation and therapeutic GCs directly promote systemic osteoporosis and muscle wasting, the mechanisms whereby they achieve this are distinct. Importantly, their interactions in vivo are greatly complicated secondary to the directly opposing actions of GCs on a wide array of pro-inflammatory signalling pathways that underpin catabolic and anti-anabolic metabolism. Several clinical studies have attempted to address the net effects of therapeutic glucocorticoids on inflammatory bone loss and muscle wasting using a range of approaches. These have yielded a wide array of results further complicated by the nature of inflammatory disease, underlying the disease management and regimen of GC therapy. Here, we report the latest findings related to these pathway interactions and explore the latest insights from murine models of disease aimed at modelling these processes and delineating the contribution of pre-receptor steroid metabolism. Understanding these processes remains paramount in the effective management of patients with chronic inflammatory disease.

## 1. Glucocorticoids and Therapeutic Glucocorticoid Excess

Synthetic glucocorticoids (GCs), such as dexamethasone, prednisolone and hydrocortisone, are widely utilized in the treatment of chronic inflammatory diseases such as chronic obstructive pulmonary disease (COPD), inflammatory bowel disease (IBD) and rheumatoid arthritis (RA), with approximately 1% of the adult population in the U.K. and U.S. receiving this class of drugs. Their mechanisms of action are diverse, with GCs suppressing a range of pro-inflammatory pathways including p38-mitogen activated protein kinases (p38-MAPK), nuclear factor kappa-light-chain-enhancer (NF-κB) and activator protein (AP-1), in addition to inducing pro-resolving factors such as glucocorticoid induced leucine zipper (GILZ) and annexin-1 [1,2,3]. These significantly reduce leukocyte infiltration at sites of inflammation, suppress the production of pro-inflammatory cytokines and chemokines and support resolution of inflammation and tissue remodelling [4,5]. Despite the potent immune-modulatory anti-inflammatory actions of therapeutic GCs, their clinical application is limited due to severe systemic side effects. These occur in up to 70% of patients and can include muscle wasting and GC induced osteoporosis (GIO) [6,7,8,9,10]. The actions of GCs on bone and muscle metabolism are well established, but themselves complicated in the backdrop of chronic inflammation by separate inflammatory driven muscle wasting and bone loss. The inflammatory pathways that mediate bone and muscle loss in chronic inflammation are in turn suppressed by the anti-inflammatory actions of GCs, further complicating the prediction of their outcome on the musculoskeletal system. Understanding the complex interactions between GC and inflammatory regulation of bone and muscle metabolism remains paramount in the effective management of patients with chronic inflammatory disease. In this review, we explore how inflammatory drivers and therapeutic GCs interact to regulate bone and muscle metabolism and consider the role of local steroid metabolism in shaping these processes.

## 2. Glucocorticoid Signalling and Regulation of Inflammation

Lipophilic GCs readily diffuse across cell membranes, signalling through the cytoplasmic GC receptor (GR) superfamily, encoded by the NR3C1 gene. Classically, GC signalling and GR transactivation occur through ligand binding of the GRα homodimer. In its unbound state, GRα forms a multi-protein complex with chaperone proteins such as heat shock proteins (HSPs), HSPp-70, HSP90 and FK506 binding protein 52 that block their nuclear localization signal (NLS) and prevent translocation to the nucleus from the cytoplasm [11]. Upon GC binding, the GRα undergoes a conformational change, allowing dissociation of chaperone proteins. Homo-dimerization and exposure of the NLS are required for nuclear translocation of the ligand bound GR, where it can influence gene expression [12] (Figure 1). This is an oversimplified view of GC signalling, as several studies utilizing the GR^dim^ mouse (possessing a mutation preventing GR dimerization) reveal that the anti-inflammatory properties of therapeutic GCs are mediated by both homo-dimeric GRα complexes and monomeric GRα to facilitate transactivation or transrepression of pro-inflammatory genes [5,13,14,15,16]. Whilst the mechanisms that underpin GR signalling have been reviewed extensively elsewhere, several key pathways are prominent in mediating the anti-inflammatory actions of GCs [17]. These include the direct GRα homodimer transactivation of anti-inflammatory genes such as secretory leukocyte protease inhibitor (*SLPI*), *MAKP-1*, *GILZ* and tristetraprolin (*TTP*), which suppress the NF-kB and p38-MAPK inflammatory pathways, in addition to the inhibition of pro-inflammatory transcription factors via their tethering to the GC bound GR [18,19,20,21]. In particular, GCs act via the GR to suppress the NF-κB and p38-MAPK inflammatory pathways and AP-1 pro-inflammatory pathways, which regulate the transcription of various genes relating to inflammation such as tumour necrosis factor-alpha (TNF-α) and interleukin-1 (IL-1β) and -6 [22]. Many of these inflammatory pathways considered are direct contributors to the process of inflammatory bone and muscle wasting and are themselves opposed by the actions of therapeutic GCs. This review will now consider how inflammation and GCs influence bone and muscle metabolism, both in isolation and in concert with one another.

## 3. Bone Metabolism

Bone metabolism is a tightly regulated process that ensures homeostasis between bone resorption and bone formation. This process maintains a balance in calcium and phosphate mineral homeostasis, as well as allowing constant healthy remodelling to compensate for external loading stress and damage and requires the close interaction between osteocytes, bone lining cells, bone forming osteoblasts and bone resorbing osteoclasts [23]. Here, in quiescent bone, osteocytes produce factors such as transforming growth factor-β (TGF-β), sclerostin and dickkopf WNT signaling pathway inhibitor 1 (DKK-1), which inhibits osteoclast and osteoblast maturation and differentiation [24]. Signals such as bone matrix damage or immobilization result in osteocyte apoptosis, leading to a removal of the inhibitory signals and increases in factors that promote osteoclastogenesis, such as macrophage colony stimulating factor (M-CSF) and receptor activator of nuclear factor kappa-β ligand (RANKL) [25,26,27]. Together, these promote osteoclast differentiation from hematopoietic precursors and increase receptor activator of nuclear factor kappa-β (RANK) pathway activation, driving multinuclear polykaryon formation and the formation of mature osteoclasts that express osteoclast specific genes including tartrate-resistant acid phosphatase (*TRAP*) and cathepsin K [28,29,30]. Mature osteoclasts form tight integrin junctions on mineralized bone matrix, forming an acidified resorption compartment that facilitates the degradation of the inorganic hydroxyapatite component of the bone [31,32]. The organic component of bone can then be degraded by lysosomal enzymes, such as cathepsin K. In parallel to this process, a reduction in factors that suppress osteoblast differentiation (such as sclerostin and DKK-1) and an increase in factors that induce osteoblast differentiation (such as WNT, TGFβ and insulin-like growth factor 1 (IGF-1)) promote the formation of osteoblast pre-cursors from mesenchymal derived progenitors [33,34,35]. This process is tightly regulated through the master transcriptional regulator runt-related transcription factor 2 (RUNX2), mediating the expression of osteoblast specific genes such as osteocalcin, osteopontin and bone sialoprotein [36,37]. As osteoblasts continue to mature, RANKL levels (which maintain osteoclasts) decrease, whilst osteoprotegerin (OPG) (the dummy receptor for RANKL that suppresses RANK signalling) increases. Together with increasing TGFβ signalling, the decrease in RANK/RANKL signalling leads to reduced osteoclast differentiation, activity and survival [38]. The transition toward the reversal phase is characterised by an increase in mature osteoblasts at the vacated osteoclast lacunae site of bone resorption. One key cell type that appears to facilitate this transition appears to be a unique cell population known as reversal cells, which cover the eroded bone surface. Here, one study has revealed that the disruption of these cells results in a loss of the initiation of bone resorption, highlighting their importance in this process [39]. Mature osteoblasts then secrete factors required for osteoid formation including organic matrix rich in type 1 collagen, osteocalcin and bone sialoprotein (BSP) [40]. This is then mineralized via the deposition of hydroxyapatite crystals, created by the flux of calcium and phosphate ions within vesicles that are deposited as a mineralized nodule, in a process that has been shown to require the enzyme alkaline phosphatase to release phosphate ions [41,42]. Ultimately, bone formation ceases as osteoblasts undergo apoptosis or are incorporated into the osteocyte network.

## 4. Regulation of Bone Metabolism by Inflammation

In diseases such as RA, IBD and COPD, ongoing systemic inflammation results in inflammatory osteoporosis, with localized destruction of bone at sites of inflammation in diseases such as RA [43,44,45,46,47]. Systemic bone loss is characterized by a general decrease in bone mineral density (BMD) at the femoral neck, hip and spine in patients, resulting in increased fracture rates [47,48,49,50]. It is widely accepted that this inflammatory bone loss results from an imbalance in the bone remodelling cycle, shifting towards resorption and away from formation [51]. Studies exploring inflammatory bone loss are complicated by immobility in patients and the impact of concurrent anti-inflammatory drugs that can influence bone metabolism. However, significant insights have been derived from in vitro studies and clinical studies.

A prominent mechanism associated with a shift toward inflammatory bone loss is the interaction of the activated immune system with bone resorbing osteoclasts. Here, changes in the inflammatory cytokine profiles in patients with chronic inflammation result in increased levels of pro-osteoclastogenic mediators and a decrease in anti-osteoclastogenic mediators. Many of the pro-osteoclastogenic cytokines upregulated in chronic inflammation, including TNF-α, IL-1, IL-6, IL-8 and IL-17, mediate their actions via an upregulation of RANKL on fibroblasts and osteoblasts, which in turn promotes osteoclastic bone resorption [52,53,54,55]. In particular, combinations of cytokines including TNF-α, IL-1 and IL-6, act synergistically to increase RANKL in inflammation [56]. Activated Th17 and B cells also upregulate *RANKL* expression promoting resorptive bone lesions in patients and in vitro in a RANKL dependent manner [57,58,59]. A recent study identified a novel cytokine induced in response to TNF-α in T cells, known as secreted osteoclastogenic factor of activated T cells (SOFAT), which has the ability to cause osteoclastogenesis in a RANKL independent manner and may have implications in bone loss induced by chronic inflammatory disease [60].

Of particular interest, TNF-α also has effects on the bone forming ability of osteoblasts in inflammation. TNF-α treatment of osteoblasts’ precursors inhibits their differentiation by suppressing the DNA binding ability of RUNX2, leading to inhibition of alkaline phosphatase expression and matrix deposition [61]. The pro-apoptotic properties of TNF-α on osteoblasts has also been observed [62]. Similarly, IL-6 treatment of osteoblasts leads to reductions in alkaline phosphatase activity and in the expression of RUNX2 and osteocalcin, with mineralisation dramatically reduced in a dose dependent manner [63]. The prominent role of the inflammatory activation of osteoclastogenesis was derived from murine models using the TNF-tg mouse of chronic polyarthritis and inflammatory bone loss. Here, blockade of both the TNF-α and the RANKL/RANK signalling pathways using anti-TNF therapy in combination with anti-osteoclastic (OPG) was able to prevent inflammatory bone erosions [64]. Bone repair was then augmented through the addition of the pro-osteoblastic hormone parathyroid hormone (PTH). These results highlight the importance of bot inflammatory activation of osteoclasts and suppression of osteoblasts in mediating systemic and localized bone loss in chronic inflammation. Consequently, these results indicate that repair of bone erosions requires a therapy that simultaneously controls inflammation while also impacting both osteoclastic bone resorption and osteoblastic bone formation to shift the balance in bone homeostasis and promote normal repair and recovery of bone.

## 5. Effects of Glucocorticoids on Bone Metabolism

Whilst GCs are widely used in the treatment of chronic inflammation, they are themselves associated with an increased risk of fractures and osteoporosis at therapeutic doses resulting in GIO. GIO is the most common form of secondary osteoporosis with risk of fracture increasing dramatically within three to six months of starting GC therapy [65]. Interestingly, these changes are reversed rapidly upon cessation of GCs, indicating a rapid and acute nature of action at the cellular level. The mechanism that underpins this appears to be primarily mediated by a substantial inhibition of osteoblastic bone formation [66]. Under physiological conditions, GCs promote osteoblast maturation. However, at higher therapeutic doses, GCs downregulate WNT agonists and upregulate WNT inhibitors, which induce apoptosis and suppress osteoblast differentiating [67,68,69]. In one clinical study examining children receiving exogenous glucocorticoids, serum levels of the WNT signalling inhibitor DKK-1 were shown to be significantly elevated, suggesting it may play a key role in reduced bone formation in GIO [70]. In studies using transgenic mice with osteoblast targeted disruption of glucocorticoid signalling, GC signalling via the GR was shown to mediate reduced bone formation through the suppression of osteoblast differentiation via the WNT pathway and through inducing osteoblast apoptosis, with animals with GR signaling disruption being protected from GC induced bone loss [67,71].

The impact of GCs on osteoclasts is less clear. Studies have reported that GC treatment results in a decrease in osteoclast number, but an increase in osteoclast longevity, potentially mediated via a GC induced increase in M-CSF production [66,72,73]. In addition, studies have shown conflicting results on the expression of osteoclastic genes in response to GCs. One study showed that dexamethasone treatment of murine calvarial bones resulted in increased mRNA levels of *Rank* and *Rankl*, leading to increased markers of osteoclast activation [74]. Other studies showed that OPG levels are suppressed or reported no change at all in RANKL and OPG levels [72,75,76]. Some insight comes from one study in children receiving exogenous GCs, where serum levels of RANKL were elevated and OPG suppressed [77]. In these patients, spontaneous osteoclastogenesis in vitro was apparent in monocytic cell precursors. Certainly, one study utilizing a murine model of therapeutic GC delivery revealed that the targeting of osteoclasts using bisphosphonates was an effective strategy to prevent both cortical and trabecular bone loss [78]. There is some evidence to indicate that the responsiveness of osteoclasts to GCs is highly dependent on the stage of cell differentiation, but these findings require further investigation [79]. The variation in GC dose, the method of administration and the models employed may explain the variation in the results reported to date, whilst their interactions with inflammatory mediators in patients with chronic inflammation should also be taken into account when investigating their bone related effects.

## 6. Glucocorticoids, Inflammation and Bone Homeostasis

Glucocorticoids directly oppose a wide array of the pathways that drive inflammatory bone loss. Amongst these, their suppression of pro-inflammatory factors such as RANKL, TNF-α and IL-6 appears to be prominent in mediating their bone sparing effects in chronic inflammatory joint destruction, through the direct suppression of osteoclastogenesis and osteoclast activation [80] (Figure 2). In contrast, their potent suppression of anabolic bone formation by osteoblasts may synergize with the deleterious actions of inflammation on osteoblasts. Consequently, the net balance of GCs on bone metabolism in the context of chronic inflammation is less clear. Several clinical studies shed light on the balance between beneficial and detrimental actions of GCs on bone metabolism in chronic inflammation. These include a study reporting no differences in BMD loss in RA patients receiving therapeutic GCs in combination with traditional disease modifying anti-rheumatic drugs (DMARDS), relative to a matched control cohort [81]. Of particular interest were studies exploring whether GCs at lower therapeutic doses might promote positive anti-inflammatory actions without eliciting detrimental bone loss. These studies reported that low dose GC therapy in RA did not increase the risk of generalized osteoporosis at the spine and hip [82,83]. Another study found that patients receiving GCs in combination with anti-TNF therapy had a 2.5% increase in BMD at the femoral neck compared to a 0.7% decrease in BMD in those using anti-TNF alone, suggesting that GCs may increase bone metabolism in this context [84]. In contrast, two further studies found that GCs’ use was associated with decreased BMD in RA patients [43,85]. Similarly, in juvenile chronic arthritis (JCA), two studies found that GC treated patients had significantly less trabecular bone and higher risk of vertebral collapse than a matched control cohort [86,87]. These studies found strong links with the dose of steroid applied, but were further complicated by the application of GCs in the developing skeleton of younger patients, who may be more vulnerable to the anti-anabolic actions of GCs than adults. The conflicting nature of these results may stem from a variety of issues, including differences in disease pathophysiology, disease activity, duration and variations in the delivery and dose of GC therapy. In addition, concomitant use of alternative therapies such as anti-TNF treatments causes further complications, making it difficult to dissect the contribution of GCs to changes in bone metabolism in chronic inflammatory disease.

## 7. Muscle Mass Related Metabolism

Similar to bone, muscle metabolism is tightly regulated to ensure a balance between anabolic and catabolic processes governing muscle mass. Its regulation is critical not only to facilitate mechanical locomotion, but also as a key site for whole body energy metabolism and homeostasis [88]. Several critical anabolic and catabolic signalling pathways determine muscle protein synthesis, muscle proteolysis and myogenesis as cellular processes in control of muscle mass.

IGF-1 has been identified as a critical factor mediating the regulation of anabolic and catabolic muscle homeostasis in adult myofibers. Produced primarily in the liver, its binding to the IGF-1 receptor (IGF1R) in skeletal muscle allows recruitment of the insulin receptor substrate 1 (IRS-1) and activation of phosphatidylinositol-3-kinase (PI3K) and phosphorylation of protein kinase B (nnown as AKT), [89,90]. Together, these result in the activation of the mammalian target of rapamycin (mTOR) signalling pathway, which results in suppression of proteolysis and activation of muscle protein synthesis. mTOR activation suppresses proteolytic, forkhead box class O family member proteins (FOXOs) and glycogen synthase kinase-3 beta (GSK-3β) pathways [91,92]. The activation of mTOR signalling promotes muscle protein synthesis through the downstream phosphorylation and inactivation of eIF4E-binding protein 1 (4E-BP1) and activation of the ribosomal protein S6 kinase beta-1 (p70S6K) [93,94,95]. When active, 4E-BP1 operates by suppressing the eukaryotic translation initiation factors (elF), which are a central rate limiting step in the regulation of protein synthesis in muscle. Here, eIF4F (a complex of initiation factors, eIF4e, eIF4G and eIF4A), promotes the translation of mRNA coding for muscle proteins by facilitating the cap dependent binding of messenger RNA to the 40S ribosomal subunit [96]. The repressor protein 4E-BP1 is a powerful negative regulator of eIF4F mediated protein translation, whilst its phosphorylation causes its dissociation from eIF4E and enables mRNA translation of anabolic muscle proteins. A second key stage in the regulation of anabolic protein metabolism in muscle occurs through the regulation of phosphorylated p70S6K by mTOR, which facilitates ribosomal biogenesis and translation capacity required for muscle protein anabolism [97]. An additional modulator of skeletal muscle mass downstream of the IGF-1/AKT pathway is GSK-3β. This protein kinase is a negative regulator of the translation initiation factor eIF2B and is phosphorylated and inactivated by AKT, allowing initiation of mRNA translation [98,99,100]. Together, the activation of these pathways by IGF-1 or insulin promote protein synthesis in muscle, favouring increased muscle mass.

The regulation of muscle catabolism shares many of these pathways and involves their inverse activation state. Proteolysis of skeletal muscle proteins through their targeted degradation by the ubiquitin-proteasome system (UPS) and autophagy pathways is under stringent control of the PI3K/AKT and mTOR signalling pathways [101]. Here, a reduction in anabolic factors such as IGF-1 or an increase in negative regulators such as myostatin, TGFβ or FGF results in a decrease of the PI3K/AKT and mTOR signalling. As AKT and mTOR kinase activity is responsible for inhibitory phosphorylation of the FOXOs, including FOXO1, FOXO3 and FOXO4 [102,103], the lack thereof allows their nuclear translocation. FOXO transcription factors bind to promoter and enhancer regions of target genes such as the E3 ligases, Atrogin-1 and muscle RING-Finger protein-1 (MURF-1) and the autophagy-related genes *LC3* and *Bnip3* [104,105,106,107]. In addition to FOXO, increased GSK-3β secondary to reduced IGF-1/AKT signalling has also been implicated in upregulating Atrogin-1 and MURF-1 [108].

The E3 ligases are the largest family of ubiquitination factors targeting muscle proteins for degradation by the UPS [109,110] and can be highly upregulated in catabolic conditions. These include the muscle specific F-box protein Atrogin-1 encoded by the *FBXO32* gene and MURF-1 encoded by the *TRIM63* gene [111,112]. Their expression is elevated in a plethora of skeletal muscle atrophy models, including immobilisation, denervation, cancer, starvation and diabetes [111,112,113]. Atrogin-1 has been shown to ubiquitinate desmin and vimentin, muscle proteins essential to sarcomere Z-disk architecture [114]. In addition, Atrogin-1 stimulates the degradation of transcription factor EIF3F, leading to impaired muscle protein synthesis [115]. This E3 ligase has also been shown to play a pivotal role in repressing myogenesis through the ubiquitination of myoblast determination protein 1 (MYOD) [116].

MURF-1 encodes a protein containing a RING finger domain, which is responsible for its ubiquitin-ligase activity [112,117]. MURF-1 ubiquitinates and catalyses the degradation of contractile proteins and thick filaments, such as myosin and troponin I, with the sparing of thin filaments such as actin [118,119]. Besides a role in the UPS, increased FOXO activation also upregulates protein degradation and clearance through the autophagy pathways [106]. In muscle, this appears to be mediated through the direct upregulation of autophagy genes such as *LC3, BNIP3* and *ATG* through the FOXO pathway during muscular atrophy [120,121,122].

Postnatal myogenesis is an anabolic process important to the maintenance of muscle mass and integrity. Insulin-like growth factor 1 (IGF-1) has been shown to be a positive driver of myogenesis, whilst fibroblast growth factor (FGF), transforming growth factor β (TGF- β) and myostatin are potent inhibitors [123,124,125,126]. In addition, various secreted WNT signalling factors positively influence myogenesis. These are regulated by an array of stimuli, including exercise, nerve innervation and dietary protein intake, and are mediated through various gene regulatory networks including the T-box family, tbx6, ripply1 and mesp-ba in mesenchymal stem cell populations [127,128,129]. Ultimately, this drives the expression of myogenic regulatory factors (MRFs) such as myogenic differentiation 1 (MYOD), myogenic factor 5 (MYF5) and myogenin (MYOG), this process being in mesenchymal derived muscle progenitor cells called satellite cells [130,131,132,133]. Although some redundancy exists in their cellular function, MYF5 is mostly implicated in mediating the proliferation of satellite cells and MYOD in their differentiation into myoblasts, whilst downstream factors, including myogenin, initiate further differentiation of mature myocytes followed by the fusion and formation of mature myotubes [130,131,132] or mostly relevant for adult muscle, fusion with myofibers.

Below, we will describe how inflammation and glucocorticoids impact these regulatory processes of muscle mass metabolism, driving a shift towards anti-anabolic and catabolic protein metabolism, resulting in muscle wasting.

## 8. Effects of Inflammation of Muscle Metabolism

Inflammation is a well established driver of muscle wasting in preclinical models and strongly relates to poor prognostic outcome and increased morbidity and mortality in patients with chronic inflammatory diseases [134]. Pro-inflammatory cytokines such as TNFβα, IL-1β and IL-6, which are elevated in chronic inflammation, are themselves reported to drive proteolysis and autophagy and suppress myogenesis and protein synthesis in muscle [135,136,137,138]. Of these, TNF-α, at the apex of the inflammatory cytokine cascade in many chronic inflammatory diseases, is critical in regulating inflammatory muscle wasting. Here, its activation of the NF-kB and p-38 MAPK pathways directly induces muscle wasting through the increased expression of the E3 ligases, atrogin-1 and MURF-1 and activation of the UPS system [139,140,141,142,143]. In models of chronic inflammation, TNF-α also downregulates circulating levels of IGF-1 and the downstream PI3K/AKT/mTOR signalling pathways, whilst upregulating the catabolic FOXO pathway to suppress protein synthesis and myogenesis in muscle [141,142,144,145]. Another factor implicated in inflammatory muscle wasting is myostatin. This is also reported to be increased in chronic inflammation, where it positively correlates with markers of disease severity. Elevated myostatin downregulates PI3K/AKT/mTOR signalling, promoting muscle atrophy [146,147]. Of interest, several studies have reported elevated endogenous GC levels as being central mediators of inflammatory muscle wasting. Here, the inflammatory activation of the hypothalamic/pituitary/adrenal (HPA) axis in response results in an elevation of circulating cortisol to mediate muscle wasting [148,149,150,151]. Of note, the blockade of endogenous GC production or muscle GR signalling could reverse muscle wasting in some experimental models [152,153]. This indicates that in addition to a direct impact of inflammation on intra-cellular muscle mass regulatory processes, activation of the HPA axis as an evolutionarily conserved response to suppress systemic inflammation can result in GC driven muscle wasting as an indirect effect of inflammation on skeletal muscle.

## 9. Effects of Glucocorticoids on Muscle Metabolism

Extended exposure to therapeutic GCs results in the rapid onset of a GC induced muscle atrophy, characterised by a decrease in myogenesis and protein synthesis and an increase in proteolysis and atrophy of muscle fibres [9,154,155,156,157]. This leads to a significant decrease in muscle fibre size, with a greater degree of wasting apparent in fast-twitch or type II muscle fibres [158]. The shift towards greater catabolic loss of protein and decreased anabolic synthesis in muscle is elicited by GCs through a number of pathways, including a decrease in IGF-1 signalling and an increase in negative regulators of the mTOR pathways such as myostatin and the protein regulated in development and DNA damage response 1 (REDD1) [157,159]. Similarly, as with inflammatory pathway activation, GCs also activate the UPS and autophagy secondary to upregulation of the FOXO1 pathway [80,154,160]. In particular, the marked increase in degradation of contractile skeletal muscle proteins through the UPS system is believed to be central in GC induced muscle wasting in vivo. This is supported by several studies demonstrating the downregulation of the PI3K/AKT/mTOR signalling pathways and the upregulation of the E3 ligases Atrogin-1 and MURF-1 in response to GCs [112]. Several studies have also demonstrated a significant increase in 4E-BP1 and suppression of p70S6K in GC induced muscle atrophy, demonstrating a role for reduced protein synthesis and regeneration [157,159]. Of interest, the restoration of IGF-1 signalling can rescue GC induced myopathy in mice, demonstrating a crucial role for this growth factor in the process of GC-induced myopathy [9,154,155,156]. Glucocorticoid mediated muscle wasting has also been shown to be rescued through the in vivo deletion of myostatin, indicating that the negative regulation of the IGF-1 pathway may also be a crucial step in this process [161,162,163].

## 10. Interaction between Inflammation and Glucocorticoids in Muscle

As with bone, many of the central inflammatory pathways that induce muscle wasting, including the NF-kB and p38-MAPK pathways, are themselves suppressed by GC signalling, suggesting that therapeutic application may protect against the process of inflammatory muscle wasting. However, other elements of inflammatory muscle wasting such as the suppression of IGF-1 and induction of myostatin and FOXO1 pathway activation are common components in both inflammatory and GC induced muscle wasting (Figure 3). Understanding how these interact in vivo remains paramount in our understanding of how therapeutic GCs should be applied in the setting of chronic inflammatory disease. Some insights arise from clinical studies exploring these processes in patients with inflammatory disease receiving GCs. Of note, in inflammatory myopathies arising directly from muscle inflammation, such as with polymyositis and dermatomyositis, GCs are effective in controlling inflammation and protecting against inflammatory muscle wasting and associated weakness [164]. Similarly, therapeutic GCs are effective in preventing muscle wasting in patients with Duchenne muscular dystrophy (DMD), where progressive muscle necrosis mediates loss of muscle [165,166,167,168,169]. However, in other inflammatory diseases, where muscle wasting occurs secondarily to inflammation at a non-muscle site, the application of therapeutic GCs is strongly associated with rapid loss of muscle mass [170,171]. These findings may suggest that GCs can oppose the process of inflammatory muscle wasting when the active inflammation is confined to the muscle, but promote muscle wasting when used to manage other systemic chronic inflammatory diseases such as RA. As with GC induced osteoporosis in patients with chronic inflammatory disease, the interpretation of these findings in relation to muscle wasting is complicated by disease severity and duration and by concurrent DMARD therapies.

## 11. Insights from Murine Models of Chronic Inflammation Receiving Therapeutic Glucocorticoids

Additional insight has come from murine models of polyarthritis receiving therapeutic GCs. These are able to circumvent issues related to differences in disease activity between patients and complications arising from the various alternative anti-inflammatory drugs used to manage disease in patients. In one such study, we examined the role of the GC corticosterone, delivered as a monotherapy in the TNF-tg murine model of polyarthritis, on net bone and muscle metabolism [80]. This revealed that therapeutic doses of GCs, whilst effective at suppressing disease activity, also potently suppressed inflammatory osteoporosis and juxta articular bone loss. This confirmed that their capacity to suppress inflammatory pathways that mediate inflammatory bone loss outweighed their deleterious effects on bone metabolism. These bone sparing effects of GCs were mediated through the suppression of pro-inflammatory osteoclasts’ activation, both systemically and at sites of inflammation. However, whilst these treatments protected from inflammatory bone loss, we still observed a suppression of anabolic bone formation in all mice receiving GCs, suggesting that long term administration may still ultimately result in GIO.

Unlike bone, therapeutic GCs markedly exacerbated muscle wasting in mice with chronic inflammation. This was characterized by a marked activation of the catabolic FOXO1 and UPS pathways [80]. Similar findings had been reported in rats, where dexamethasone exacerbated inflammatory muscle wasting in models of sepsis [172]. These data indicate that the beneficial effects of inflammatory suppression by GCs in muscle were not sufficient to outweigh their deleterious actions on muscle metabolism. Whilst further work is required to better elucidate the actions of therapeutic GCs in the context of chronic inflammation, these data shed light on the potential strengths and weaknesses of their application in muscle and bone. In particular, they indicate that the management of side effects in muscle may need to be prioritised over those in bone, in patients with chronic inflammatory diseases receiving therapeutic GCs.

## 12. Pre-Receptor Regulation of Therapeutic GC Action to Protect against Side Effects

The pre-receptor metabolism of GCs by the 11β-hydroxysteroid dehydrogenase (11β-HSD) enzymes is recognised as a critical step in mediating GC signalling in many peripheral tissues. These are the 11β-HSD type 1 (11β-HSD1) and type 2 (11β-HSD2). 11β-HSD1 is expressed in many tissues, including the liver, bone, muscle and fat, where it converts inactive endogenous and therapeutic GCs (such as corticosterone and prednisone) to their active counterparts (such as cortisol and prednisolone), leading to a local accumulation and concentration of active GCs [173,174,175] (Figure 4). In contrast, 11β-HSD2 solely inactivates endogenous and therapeutic GCs within the kidney, providing circulating inactive GC substrate for the peripheral 11β-HSD1 enzyme and supporting renal clearance of GCs [175]. Several key studies have demonstrated a critical role for the pre-receptor activation of GCs by 11β-HSD1 in mediating the deleterious actions of therapeutic GCs in muscle and bone [176,177]. Here, animals with transgenic deletion of 11β-HSD1 are resistant to both exogenous GC induced muscle wasting and osteoporosis. This raises the exciting possibility that therapeutic 11β-HSD1 inhibitors, widely explored in the management of metabolic disease, may prevent bone loss and muscle wasting in patients with chronic inflammatory diseases receiving GCs [178,179]. Further studies lend strength to this concept, showing that 11β-HSD1 is potently upregulated within muscle cells and osteoblasts, where it is potently upregulated by circulating inflammatory cytokines such as TNF-α and IL-1β [174,180,181]. Despite this, caution should be applied in this context, given that systemic deletion of 11β-HSD1 can exacerbate disease activity in murine models of inflammation, secondary to a reduction in reactivation of endogenous GC at sites of inflammation [182,183]. Consequently, further studies are required to delineate the potential benefits and risks of 11β-HSD1 inhibition in chronic inflammatory disease.

## 13. Conclusions

Both chronic inflammation and therapeutic GCs are potent drivers for systemic bone and muscle wasting resulting from an imbalance between anabolic and catabolic homeostasis. Whilst therapeutic GCs oppose many of the inflammatory pathways that drive bone and muscle wasting, they share common pathways that promote anti-anabolic and catabolic metabolism of bone and muscle and can drive or exacerbate these deleterious processes in chronic inflammatory disease. However, these relationships are invariably complicated by the nature of the inflammatory disease in which therapeutic GCs are utilized. Intriguingly, 11β-HSD1 inhibitors may possess the potential to prevent the deleterious actions of therapeutic GCs in the backdrop of chronic inflammation. However, further studies are required to assess their efficacy and safety in this context.

## Figures and Tables

**Figure 1 ijms-20-05768-f001:**
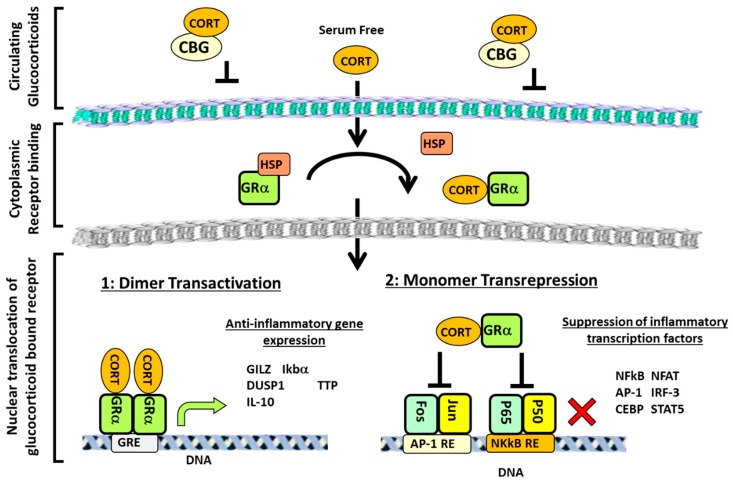
Overview of glucocorticoid (GC) signalling pathways. The majority of glucocorticoids (GCs) in the circulation are bound by corticosteroid-binding globulin (CBG), which prevents diffusion across the membrane. However, free GCs can readily enter the cell, where they bind to the GR in the cytoplasm. This induces a conformational change in the glucocorticoid receptor (GR), which causes the dissociation of chaperone molecules, such as heat shock proteins (HSPs), to expose the nuclear localisation signal (NLS) and allow translocation of the GC/GR complex to the nucleus. Here, the GR can either dimerise to transactivate anti-inflammatory genes or signal as a monomer to inhibit pro-inflammatory transcription factors. Cortisol (CORT), nuclear factor of activated T-cells (NFAT), CCAAT-enhancer-binding proteins (or C/EBPs), nuclear factor kappa-light-chain-enhancer of activated B cells (NF-κB), p38 mitogen-activated protein kinases (p-38-MAPK), glucocorticoid induced leucine zipper (GILZ), secretory leukocyte protease inhibitor (SLPI), tristetraprolin (TTP), mitogen-activated protein kinase-1 (MKP-1), activator protein 1 (AP-1), signal transducer and activator of transcription 5 (STAT5), and response element (RE).

**Figure 2 ijms-20-05768-f002:**
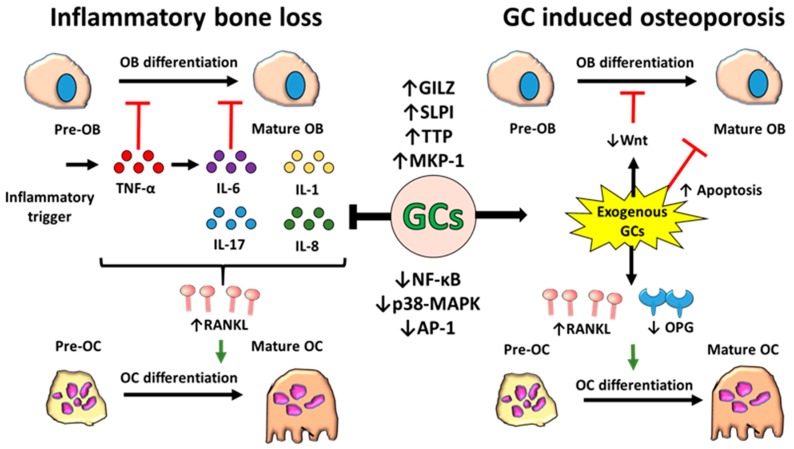
Schematic representation of the effects of inflammation and glucocorticoids (GCs) on bone remodelling. During inflammation, elevated levels of pro-inflammatory cytokines, such as TNF-α and IL-6, inhibit the differentiation of bone forming osteoblasts from their precursors. These cytokines, along with other pro-inflammatory mediators including IL-1, IL-17 and IL-8, also upregulate the expression of receptor activator of nuclear factor kappa-Β ligand (RANKL), which binds to receptor activator of nuclear factor kappa-Β (RANK) on pre-osteoclasts and triggers their differentiation into mature bone resorbing osteoclasts. Overall, bone formation is decreased while bone resorption is increased, leading to a net loss of bone. Although GCs suppress inflammation via suppression of pro-inflammatory factors and induction of anti-inflammatory mediators, they can also independently drive bone loss by inhibiting differentiation and inducing apoptosis of osteoblasts whilst increasing osteoclast differentiation by stimulating expression of RANKL and decreasing its decoy receptor osteoprotegerin (OPG). Osteoblasts (OBs), p38 mitogen-activated protein kinases (p-38-MAPK), glucocorticoid-induced leucine zipper (GILZ), secretory leukocyte protease inhibitor (SLPI), tristetraprolin (TTP), mitogen-activated protein kinase-1 (MKP-1), activator protein 1 (AP-1), OC (osteoclast), canonical WNT signalling (WNT), and nuclear factor kappa-light-chain-enhancer of activated B cells (NF-κB).

**Figure 3 ijms-20-05768-f003:**
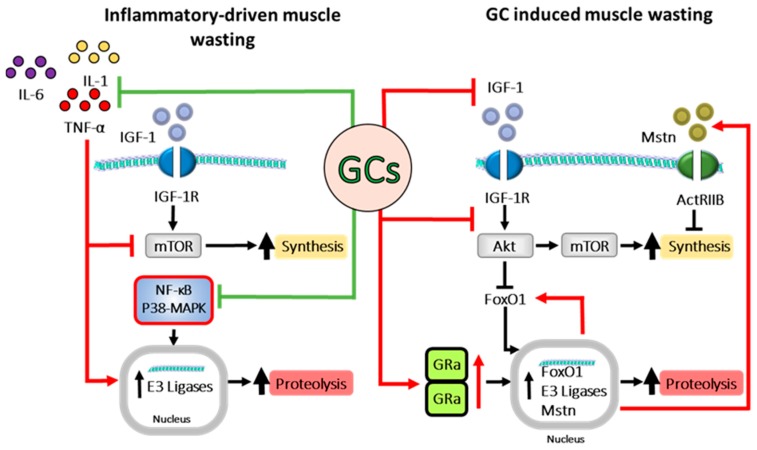
Schematic representation of signalling pathways involved in both inflammatory driven and GC induced muscle wasting and their interactions. Inflammatory cytokines such as TNF-α and IL-1 inhibit mammalian target of rapamycin (mTOR) signalling, dampening muscle protein synthesis, whilst simultaneously inducing transcription of E3 ligases, leading to muscle proteolysis. Glucocorticoids (GCs) inhibit inflammatory signalling, including nuclear factor kappa-light-chain-enhancer of activated B cells (NF-κB) signalling, and therefore decrease inflammatory driven muscle wasting. Despite this, GCs also drive muscle wasting through several pathways themselves, including suppression of the IGF-1/AKT/mTOR signalling cascade, leading to decreased protein synthesis and increased FOXO1 transcription. GR activation and dimerization induce the transcription of myostatin (MSTN), FOXO1 and other E3 ligases, leading to increased proteolysis and diminished protein synthesis. Forkhead box protein O1 (FOXO1), mammalian target of rapamycin (mTOR), insulin like growth factor (IGF-1), (p-38-MAPK), glucocorticoid induced leucine zipper (GILZ), secretory leukocyte protease inhibitor (SLPI), tristetraprolin (TTP), nuclear factor kappa-light-chain-enhancer of activated B cells (NF-κB), glucocorticoid receptor (GR), myostatin (MSTN), and IGF-1 receptor (IGF-1 R).

**Figure 4 ijms-20-05768-f004:**
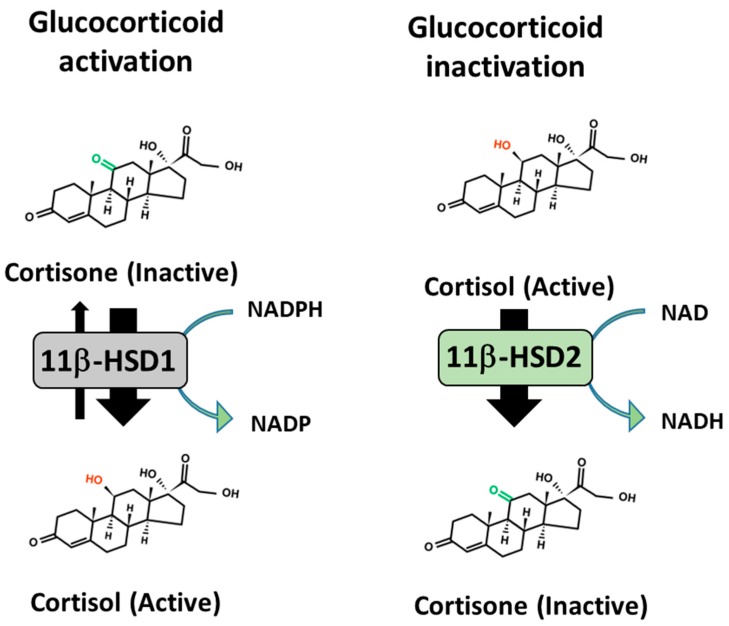
Pre-receptor metabolism of GCs by 11β-HSD1. 11β-hydroxysteroid dehydrogenase (11β-HSD) type 1 is a bidirectional enzyme that predominantly reduces inactive GCs to their active counterparts in an NADPH dependent manner, whilst 11β-HSD type 2 is an NAD+ dependent unidirectional enzyme that converts active GCs to their inactive counterparts.

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
