# Peer review of "Exploring the Interface between Inflammatory and Therapeutic Glucocorticoid Induced Bone and Muscle Loss"

_ijms, 2019, doi:10.3390/ijms20225768_

Round 1
Reviewer 1 Report
The manuscript « Exploring the Interface Between Inflammatory and Therapeutic Glucocorticoid Induced Bone and Muscle Loss » reviews how chronic inflammation and its treatment with Glucocoticoids (GCs) impact on bone remodeling and muscle metabolism, leading to bone and muscle loss.
Excellent review: well written and particularly interesting on numerous points such as the one on the interaction between the immune system and osteoclasts, with regard to the factor secreted by SOFAT or the one on the TNF-tg mouse model for chronic polyarthritis.
I suggest including one interesting study on pamidronate treatment in Mdx mice treated with GCs. Pre-treatment with Pamidronate Improves Bone Mechanical Properties in Mdx Mice Treated with Glucocorticoids. Chen J, Yoon SH, Grynpas MD, Mitchell J. Calcif Tissue Int. 2019 Feb;104(2):182-192. doi: 10.1007/s00223-018-0482-5. Epub 2018 Oct 9.
Page 3 in “3.Bone metabolism”, I suggest also including one of Delaisse JM’s studies that describe the local coupling between bone resorption and formation in basic multicellular units with osteoclasts and osteoblasts, covered by canopies (e.g. Andersen TL, Abdelgawad ME, Kristensen HB, Hauge EM, Rolighed L, Bollerslev J, Kjærsgaard-Andersen P, Delaisse JM. Am J Pathol. 2013 Jul;183(1):235-46. doi: 10.1016/j.ajpath.2013.03.006. Epub 2013 Jun 6.)
Minor corrections:
Human gene or protein symbol should be capitalized and genes or mRNA name should be in italics, but proteins should not be in italics. (g. HSP for the protein and HSP for the gene or mRNA). Please check : “hsp” page 2 line 14, page 3 line 5; wnts/wnt page 3 line 28 page 5 lines 11 and 14; RANK and RANKL (mRNA) page 5 lines 21 and 22… Indeed if the gene/mRNA (italized) and protein (no-italized) format is maintained correctly throughout the text, it would help the reader to know whether “all RANKL and OPG levels” as mentioned page 5 line 23, correspond to a protein or mRNA increase. Please see Guidelines for Human Gene Nomenclature. Full name for abbreviation is needed in figure legends. Please specify CORT in figure 1 legend.
Author Response
We would like to thank the reviewers for their insightful comments relating to our manuscript. We have added in the key manuscripts highlighted in these comments and discussed them accordingly as follows:
Reviewer 1:
Excellent review: well written and particularly interesting on numerous points such as the one on the interaction between the immune system and osteoclasts, with regard to the factor secreted by SOFAT or the one on the TNF-tg mouse model for chronic polyarthritis.
I suggest including one interesting study on pamidronate treatment in Mdx mice treated with GCs. Pre-treatment with Pamidronate Improves Bone Mechanical Properties in Mdx Mice Treated with Glucocorticoids. Chen J, Yoon SH, Grynpas MD, Mitchell J. Calcif Tissue Int. 2019 Feb;104(2):182-192. doi: 10.1007/s00223-018-0482-5. Epub 2018 Oct 9.
Thank you for highlightinhg this highly pertinent literature. We have included a bried summary of these findings in the context of GC action on bone as follows “Certainly, one study utilizing a murine model of therapeutic GC delivery, revealed that the targeting of osteoclasts using bisphosphonates is an effective strategy to prevent both cortical and trabecular bone loss (75).”
Page 3 in “3.Bone metabolism”, I suggest also including one of Delaisse JM’s studies that describe the local cou"pling between bone resorption and formation in basic multicellular units with osteoclasts and osteoblasts, covered by canopies (e.g. Andersen TL, Abdelgawad ME, Kristensen HB, Hauge EM, Rolighed L, Bollerslev J, Kjærsgaard-Andersen P, Delaisse JM. Am J Pathol. 2013 Jul;183(1):235-46. doi: 10.1016/j.ajpath.2013.03.006. Epub 2013 Jun 6.)
We have addressed this as follows: “The transition toward the reversal phase is characterised by an increase in mature osteoblasts at the vacated osteoclast lacunae site of bone resorption. One key cell type that appears to facilitate this transition appears to be a unique cell population known as reversal cells, which cover the eroded bone surface. Here, one study has revealed that the disruption of these cells, results in a loss of the initiation of bone resorption, highlighting their importance in this process (39).”
Minor corrections:
Human gene or protein symbol should be capitalized and genes or mRNA name should be in italics, but proteins should not be in italics. (g. HSP for the protein and HSP for the gene or mRNA). Please check : “hsp” page 2 line 14, page 3 line 5; wnts/wnt page 3 line 28 page 5 lines 11 and 14; RANK and RANKL (mRNA) page 5 lines 21 and 22… Indeed if the gene/mRNA (italized) and protein (no-italized) format is maintained correctly throughout the text, it would help the reader to know whether “all RANKL and OPG levels” as mentioned page 5 line 23, correspond to a protein or mRNA increase. Please see Guidelines for Human Gene Nomenclature. Full name for abbreviation is needed in figure legends. Please specify CORT in figure 1 legend.
These have been ammended
Reviewer 2 Report
The review is interesting and well written only few items should be adjusted:
- the following two papers should be added and discussed about the role of RANKL-OPG and Wnt inhibitors in GIO:
J Clin Endocrinol Metab. 2009 Jul;94(7):2269-76.
Am J Physiol Endocrinol Metab. 2013 Mar 1;304(5):E546-54.
- pag 4 line 45: "were able to" is repeated 2 times
Author Response
We would like to thank the reviewers for their insightful comments relating to our manuscript. We have added in the key manuscripts highlighted in these comments and discussed them accordingly as follows:
Reviewer 2:
- the following two papers should be added and discussed about the role of RANKL-OPG and Wnt inhibitors in GIO:
J Clin Endocrinol Metab. 2009 Jul;94(7):2269-76.
We have included this important reference “In one clinical study examining children receiving exogenous glucocorticoids, serum levels of the wnt signalling inhibitor DKK-1 were shown to be significantly elevated, suggesting it may play a key role in reduced bone formation in GIO (70)”
Am J Physiol Endocrinol Metab. 2013 Mar 1;304(5):E546-54.
Thank you for highlighting this interesting paper. We have discussed these findings as follows: Some insight comes from one study in children receiving exogenous GCs, where serum levels of RANKL are elevated, and OPG suppressed (77). In these patients, spontaneous osteoclastogenesis in vitro is apparent in monocytic cell precursors
- pag 4 line 45: "were able to" is repeated 2 times
This has been amended